# Attention Deficits in Healthcare Workers with Non-Clinical Burnout: An Exploratory Investigation

**DOI:** 10.3390/ijerph21020239

**Published:** 2024-02-19

**Authors:** Sergio L. Schmidt, Bruno da Silva Cunha, Julio Cesar Tolentino, Marcela J. Schmidt, Guilherme J. Schmidt, Alice D. Marinho, Eelco van Duinkerken, Ana Lucia Taboada Gjorup, Jesus Landeira-Fernandez, Carolina Ribeiro Mello, Sarah Pini de Souza

**Affiliations:** 1Post-Graduate Program, Department of Neurology, Federal University of the State of Rio de Janeiro, Rio de Janeiro 21941-901, Brazil; slschmidt@terra.com.br (S.L.S.); juliotolentinonovo@gmail.com (J.C.T.); marcela-schmidt@hotmail.com (M.J.S.); guilhermejschmidt@gmail.com (G.J.S.); alicedmarinho@edu.unirio.br (A.D.M.); e.vanduinkerken@amsterdamumc.nl (E.v.D.); analuciataboada@yahoo.com.br (A.L.T.G.); carolina-mello.cm@ebserh.gov.br (C.R.M.); sarahgpini@gmail.com (S.P.d.S.); 2Department of Medical Psychology, Amsterdam University Medical Center, Vrije Universiteit, 1081 HV Amsterdam, The Netherlands; 3Department of Psychology, Pontifical Catholic University, Rio de Janeiro 22451-900, Brazil; landeira@puc-rio.br

**Keywords:** burnout, healthcare professionals, attention deficits, impulsivity, executive functions

## Abstract

Burnout syndrome is characterized by exhaustion, cynicism, and reduced effectiveness. Workers with high burnout scores who continue their professional activities are identified as experiencing non-clinical burnout (NCB), which includes early stages where burnout symptoms (BNS) are present but not yet severe enough to necessitate work leave. This study aimed to investigate the impact of BNS on attention performance among healthcare workers (HCWs) at a COVID-19 reference hospital during the pandemic. The Maslach Burnout Inventory (MBI) was applied to assess the three burnout dimensions. The Continuous Visual Attention Test (CVAT) evaluated four different attention subdomains. Participants were divided into two groups based on their scores on the MBI: controls and NCB. Thirteen controls were matched with 13 NCB subjects based on age, sex, and HCW category. This sample (*n* = 26, 65% male) consisted of 11 physicians and 15 nursing professionals with a mean age of 35.3 years (standard deviation = 5.47). NCB subjects had higher impulsivity than controls. There were not any significant group differences in the other attention subdomains. We found significant correlations between impulsivity and all burnout dimensions: higher absolute scores in BNS are associated with higher impulsivity. We concluded that NCB leads to executive attention deficits

## 1. Introduction

Burnout is considered a consequence of occupational chronic stress and constitutes an important health risk factor among working populations [1]. Burnout consists of three dimensions: exhaustion, cynicism, and reduced personal efficacy. According to Schaufeli and Salanova [2], exhaustion is the depletion of emotional resources, cynicism is related to distancing from work, and reduced personal efficacy is associated with low work self-esteem. Moreover, studies have reported cognitive deficits in workers diagnosed with clinical burnout [3,4,5,6,7]. In this regard, Deligkaris et al. [8] suggested that a cognitive dimension be included in the burnout definition, arguing that burnout has a significant impact on workers’ cognitive performance.

A previous study [9] proposed that burned-out individuals can be identified by a high score on the exhaustion scale combined with a highly unfavorable score on at least one of the two other main components of burnout. Working employees with high burnout scores who keep working are commonly referred to as non-clinical burnout (NCB) employees [10].

NCB represents an early stage of burnout where individuals experience symptoms of exhaustion and depersonalization but continue to work [11]. Initially, NCB manifests as exhaustion and depersonalization, with prolonged exposure to stressors leading to more severe symptoms [10]. Over time, these symptoms escalate, affecting functional abilities and transitioning into clinical burnout [10]. In its clinical form, burnout encompasses emotional and physical exhaustion and depersonalization and includes a reduced sense of personal or professional accomplishment [10]. Additionally, clinical burnout subjects are required to take leave from work [11].

Healthcare workers (HCWs) are usually subjected to a high level of stress [12,13]. High rates of exhaustion were found among HCWs before the pandemic [14,15]. During the COVID-19 pandemic, HCWs were significantly exposed to the risk of virus infection while dealing with patients. Accordingly, recent studies have suggested that exhaustion increased dramatically among HCWs during the COVID-19 pandemic [16,17,18,19,20,21]. A study in Brazil found that depersonalization and emotional exhaustion were the most affected dimensions in healthcare professionals who were working during the pandemic [22]. Therefore, the presence of burnout symptoms across different HCW roles indicated that this population could provide an adequate sample for the study of cognitive performance in subjects who keep working with high levels of burnout (NCB).

Although cognitive deficits have been described in subjects with clinical burnout, the presence of cognitive impairment among NCB populations remains a matter of controversy. Some studies have shown the presence of cognitive deficits [10] while others have failed to find such impairments [23,24]. If little is known about cognitive impairments in NCB, even less is known when NCB involves HCWs. 

However, the study of cognitive performance in HCWs who continued to work with burnout symptoms is of practical and theoretical interest. In practical terms, cognitive problems lead to work-related errors. Experiencing cognitive problems can directly affect work performance, worsening feelings of distress, depression, and anxiety [4,25,26].

In theoretical terms, some authors have suggested the presence of cognitive deficits in NCB. In contrast, others have proposed that compensatory mechanisms can obscure differences in cognitive performance between NCB and controls [27]. For those who advocate the presence of cognitive deficits in NCB, the adverse effects of burnout on cognition might be associated with a decrease of the brain-derived neurotrophic factor, which would disrupt brain functioning and lead to cognitive deficits even in the early stages of burnout [28]. Conversely, two theories are frequently mentioned to explain how NCB employers could cope with cognitive difficulties at work: cognitive reserve and self-regulation theories [29,30]. The cognitive reserve theory suggests that when NCB subjects carry out challenging tasks, they employ new brain networks based on their cognitive reserve processes, allowing them to use other cognitive strategies [29]. The self-regulation theory proposes that NCB employees might still be able to self-regulate and initiate the cognitive reserve to achieve optimal performance because they retain the ability to inhibit prepotent responses and guide their cognitive sources to attain a goal [30].

Several factors, such as the heterogeneity of cognition assessment tools, may underlie the inconclusive findings on cognitive performance and burnout symptoms in NCB subjects. In this regard, the attention domain plays a fundamental role in cognition. Attention is the ability to choose and concentrate on relevant stimuli [31]. In daily life, attention is essential for driving, learning, and performance in several jobs [32,33]. Thus, decreased attentional performance could lead to a higher risk of work-related accidents, as workers might be more prone to make mistakes [33]. Moreover, attention is crucial for adequately function in the other cognitive domains [31,34]. However, attention performance has not been systematically studied in NCB subjects. 

Converging lines of evidence suggest that the attention system can be separated into four subdomains: alertness, behavioral inhibition, focused attention, and sustained attention [35,36]. Intrinsic alertness refers to the internal control of arousal without an external cue, while behavioral inhibition is the ability to control inadequate responses. Focused attention reflects the ability to respond to correct targets, and sustained attention is frequently described as the ability to concentrate over time to detect correct events [31]. Attention subdomains can be reliably measured by Continuous Performance Tests, such as the Continuous Visual Attention test (CVAT) [37,38,39,40,41].

While previous research has acknowledged the cognitive impacts of clinical burnout, our study delves deeper into the attention subdomains in NCB. Therefore, in the present study, we conducted an exploratory investigation of the performance of HCWs working during the COVID-19 pandemic in the CVAT. Specifically, we investigated whether the performance in each attention subdomain was affected in individuals exhibiting NCB. We hypothesized that NCB leads to attention deficits in HCWs. 

## 2. Materials and Methods

### 2.1. Participants

Volunteers were recruited at a tertiary university hospital in Rio de Janeiro, Brazil, between April and December 2020, during the COVID-19 pandemic. HCWs included medical doctors, nurses, and nurse’s aides. First, we performed a brief clinical interview with HCWs aged between 25 and 45 years old. The age lower limit (25 years) was set based on a previous meta-analysis that showed higher rates of emotional exhaustion and depersonalization among younger nurses [42]. As such, younger age might have intensified the symptoms of burnout, whereas advancing age may have functioned as a mitigating factor against burnout. The upper limit of 45 years was chosen based on previous studies that have described a decline in processing speed after 45 years old [43,44,45]. Accordingly, individuals older than 45 years have higher variability in the scores of the CVAT [46], which could impact the probability of detecting group differences and consequently the need to use larger sample sizes. Then, we applied the following exclusion criteria: use of antipsychotic or antiepileptic medications that might interfere with performance on the CVAT, a history of head trauma with loss of consciousness, current use of alcohol or illicit substances, pre-existing neurological or psychiatric conditions, shift workers, and a past medical record of COVID-19. The exclusion of participants who had COVID-19 was necessary because the infection itself can cause attention impairment [39,40,41]. All workers were tested at the beginning of their work shifts to exclude fatigue. Based on the Maslach Burnout Inventory (MBI) responses, we established two groups: controls and non-clinical burnout. The classification criteria for these two groups are detailed in the classification subsection (Section 2.2) and were based on the three dimensions of the Portuguese validation of the MBI (Section 2.2 and Section 2.4.2). Finally, the selected participants performed the attention task. The experimenter who administered the attention test was blind to the participant’s group. 

As described by previous authors [7], here it was also difficult to find HCWs with high burnout symptoms who were willing to participate in the study. Thirteen participants with high burnout symptoms (non-clinical burnout group) completed the attentional test. Then, from the remaining 24 participants without burnout symptoms, we selected 13 controls paired by age, sex, educational level, and HCW role. The minimum sample size had been previously determined by a power analysis (Section 2.3).

Participation in this study was voluntary and carried out according to the recommendations of the Research and Ethics Committee of the Federal University of the State of Rio de Janeiro, Brazil (CAAE: 69406817.1.0000.5258), adhering to the Declaration of Helsinki. All subjects gave written informed consent.

### 2.2. Classification of the Participants Based on Their Maslach Burnout Inventory (MBI) Scores

Burnout can be conceptualized as a continuum, ranging from low to high degrees of the phenomenon. Higher scores in the emotional exhaustion and depersonalization dimensions, and lower scores in the personal accomplishment dimension, are indicative of greater burnout [47,48]. Some studies have chosen to consider only the emotional exhaustion and cynicism dimensions to diagnose burnout, suggesting that the ineffectiveness dimension (also known as personal accomplishment) reflects a personality pattern [49,50]. Other studies have chosen a one-dimensional criterion (emotional exhaustion), considering that cynicism and low personal achievement would be different phenomena [51]. 

To classify the subjects according to the presence of significant burnout symptoms (yes or no), we applied the most restrictive criterion as proposed by Brenninkmeijer and collaborators [9], which involves the presence of high exhaustion, high cynicism, and low personal efficacy. We used the cutoff values of the third edition of the MBI manual validated for the Brazilian population. These values are as follows: average EX ≥ 3.2, average CY ≥ 2.2, and average PE ≤ 4.0 [52,53,54,55].

Here, we also treated burnout as a continuous variable considering the values of each one of the three burnout dimensions. For each participant, we summed the scores obtained in the questions related to a particular dimension. Then, we divided the obtained sum by the number of questions specifically related to that dimension.

### 2.3. Power Analysis

To estimate the required minimum sample size, we performed a power analysis considering the two different statistical approaches outlined in Figure 1, i.e., mean differences in attention performance between two groups dichotomized according to the presence of burnout symptoms (first approach) and associations between burnout dimensions and attentional performance (second approach). Therefore, two distinct analyses were performed. However, for both conditions, α = Type I error = 0.05 and β = Type II error = 0.20 (power = 1 − β = 0.80) were applied. 

When the participants were dichotomized according to the presence or absence of burnout symptoms, we performed MANCOVAs because they would account for any potential correlations between the dependent variables (attention subdomains). Thus, hypothetically, the MANCOVAs could show significant differences between the means while the individual ANCOVAs and the *t*-tests did not. However, irrespective of the results of the MANCOVAs, we always performed post hoc *t*-tests to determine where significant group differences in attention performance existed. As all the post hoc comparisons are variations of *t*-tests, we performed power analyses considering independent *t*-tests. We estimated the minimum differences (Δ) considering that they must reach magnitude levels that have clinical significance. For each variable of the abbreviated attention test (version 1.5 min), the population standard deviation and the mean difference with a real clinical significance were estimated based on comparisons (larger samples in previous studies) between healthy controls and patients with clinically defined attention disorders. In addition, some previous investigators define objective cognitive impairment in a particular domain as a score that is 1.5 standard deviations (SD) or more below the normative mean [56]. Considering all these matters, we performed a power analysis with the lowest Cohen’s d among the four CVAT variables. For an allocation ratio of 1 and differences of 1.5 SD above the normative mean (note that in the CVAT, a higher score means a poorer performance), we found that the minimum sample size was 18 subjects, 9 in each group. Therefore, considering α = 0.05, β = 0.20, power = 0.80, and d = 1.5 the present study reached an adequate sample size for the analysis of any possible mean differences in attentional performance between the two groups (*n* = 26, 13 in each group). 

When burnout was characterized as a continuous variable, the relationships between each one of the three burnout dimensions with the variables of the CVAT (attention subdomains) were analyzed using Pearson correlation coefficients (R). In regression research, the most common effect size is the squared Pearson correlation, R^2^. Effect size is used to decide a priori what relationship should be considered for practical significance. Consequently, the first task in a sample size analysis for correlation analysis should be the identification of the magnitude of the correlation expected in the population. Unfortunately, in the present study, we had no empirical basis for any presumed population correlation (ρ). When no other rationale is available, [57] recommended that R^2^ = 0.25 should serve as an upper limit. Additionally, Cohen [58] suggested that R^2^ = 0.26 is a large regression effect. As a last resort, we here considered ρ between 0.50 and 0.55. Using these large-size effects, the minimum sample size varied between 29 and 23 subjects, respectively. Thus, our sample size (*n* = 26) was within this range.

### 2.4. Procedures

We provided a comprehensive description of all elements involved in this study, following the guidelines proposed by the ©STROBE initiative [59]. After applying exclusion and inclusion criteria (interview), the selected participants filled out the MBI. Then, participants were divided into two groups based on their scores across the inventory’s three dimensions: controls and those exhibiting significant burnout symptoms, the latter group hereafter referred to as the non-clinical burnout (NCB) group. Finally, they performed the continuous visual attention test (CVAT). The researchers who administered the CVAT were blinded to the participants’ burnout symptoms. All quantitative analyses were performed using the matched groups. 

The CVAT assessed four distinct attention subdomains, while the Maslach inventory measured three different burnout dimensions. We analyzed the mean group differences in attention performance, considering the dichotomized sample (controls vs. NCB). Additionally, within the selected sample, we treated the burnout dimensions as three continuous variables and examined any significant relationships between the scores for each burnout dimension and the performance on the attention subdomains. 

#### 2.4.1. Computerized Visual Attention Test (CVAT)

The CVAT (Figure 2) involved a continuous stream of visual stimuli appearing on a computer screen at fixed intervals. Participants were instructed to press the spacebar as swiftly as possible when a specific target stimulus appeared, while refraining from responding to non-target stimuli. The test was self-paced, with stimuli presented for 250 ms and a 750-ms interstimulus interval. Out of 90 trials, 72 (80%) were targets, and 18 (20%) were non-targets. The test provided measures of reaction time (RT) and accuracy for both target and non-target stimuli, as well as a measure of response variability which offered insights into the stability of attentional performance and is linked to sustained attention [60].

The average reaction time for correct responses (RT) was calculated for each participant. Furthermore, intraindividual reaction time variability (VRT) for all correct responses to the target was determined, estimated by the standard deviation (SD) of all correct RTs per individual. Accuracy was determined by tallying the number of omission and commission errors (OE and CE, respectively). An omission error was counted when a participant failed to respond to a target, while a commission error represented any incorrect responses to non-target stimuli.

#### 2.4.2. Maslach Burnout Inventory (MBI)

The MBI [61] is a self-report questionnaire that took approximately 10 to 15 min to complete and comprised 16 items assessing the three most important symptoms of burnout: exhaustion (sample item: ‘I feel used up at the end of the work-day’), cynicism (sample item: ‘I have become less enthusiastic about my work’), and reduced personal efficacy (sample item: ‘In my opinion, I am good at my job’). Participants indicated their frequency of experiencing each item on a 7-point scale, which ranged from “never” to “every day.” Scores for each dimension were derived by summing the scores for the respective items and then determining the average score.

### 2.5. Statistical Analysis

#### 2.5.1. Mean Differences in Attention Performance between Two Groups Dichotomized According to the Presence of Burnout Symptoms

To verify whether there were significant differences between HCWs with high burnout symptoms and those without burnout symptoms, a MANCOVA was performed including RT, VRT, OE, and CE as dependent variables, and the group (high non-clinical burnout vs. no burnout) as the independent variable. Box’s M-test was used to assess the homogeneity of the covariance matrices. A significant MANCOVA indicates that at least one dependent variable is different between the groups, thus allowing for further post hoc univariate ANCOVAs. A MANCOVA/ANCOVA approach was chosen as it has been shown to give robust results even when variables are not normally distributed [62]. To determine whether there were mean differences between the two groups, we also performed independent *t*-tests on the CVAT variables. 

In the present study, we balanced the groups considering potential confounders (age and sex). As we have a matching design, it was possible to perform the MANCOVA and respective ANCOVAs without confounders and covariates. However, matching is more complex than just balancing groups for potential confounders. Covariates and confounders are typically included in statistical models to account for the variance they explain in the dependent variables. Some investigators [63] have proposed that the benefit obtained from accounting for the variance that the confounders explain in the dependent variables is greater than the increase of power obtained without confounders. Therefore, even in a matched design sometimes it is necessary to control for the matching factors in the analysis. For this reason, we decided to perform MANCOVAs and ANCOVAs with and without confounders. Age and sex were used as covariates. We also performed the MANCOVAs and respective ANCONVAs considering OE, CE, and VRT as dependent variables and RT, age, and sex as confounders. The use of RT as a cofactor was proposed by Linden et al. [7]. These authors considered that CEs are inversely related to RTs and, thus, average RT might obscure differences in CE and VRT. 

It should be mentioned that fatigue was not considered as a confounding variable for two reasons: first, we did not include shift workers and second, the CVAT lasts only 1.5 min. 

#### 2.5.2. Correlation between Burnout Dimensions and Attention Subdomains

All of the possible associations between the three burnout variables and the CVAT variables were analyzed with Pearson correlation coefficients. Here, we checked whether the data satisfied all of the necessary assumptions of correlation linear analysis. The following assumptions were carefully analyzed: linearity (relationships of the predictors to the outcomes should be linear), normality of the errors (the residuals should be normally distributed), homoscedasticity (the variance of the residuals should be homogeneous across levels of the predicted values), and independence (the errors associated with one observation should not be correlated with the errors of any other observation). Furthermore, we also verified if any single observation or small group of observations made significant differences in the correlation coefficients (outliers and influential observations).

## 3. Results

One hundred and fifty-four HCWs were interviewed; 53 were excluded because of a previous COVID-19 infection and 22 for other medical reasons. From the remaining 79 subjects, 13 could be classified into the NCB group and 24 did not show any relevant symptom in all three burnout dimensions. The controls were matched with the 13 NCB subjects based on age, sex, and HCW category. After applying the matching criteria, 13 eligible participants without burnout symptoms were selected to be compared with 13 NCB subjects. 

For the final selected sample (*n* = 26), the age ranged from 25 to 44 years (mean = 35.31; standard deviation = 5.47), with 17 men and 9 women. The sample of health professionals consisted of 11 physicians and 15 nursing professionals. Table 1 presents demographic data and MBI results. Although the two groups were not perfectly matched, we did not find any significant group differences in the demographic data. Regarding MBI scores, because of the definition of the NCB group, significant group differences (NCB vs. controls) were found for the three burnout dimensions.

Figure 3 presents the raw data of the four CVAT variables according to the two dichotomized groups. Visual inspection indicated that CE is higher in the NCB group as compared to controls. Average reaction time tended to be higher in controls, but the amplitude of the difference is within the range of one standard error of the mean.

The MANCOVA showed a significant overall effect of non-clinical burnout on the attention test, F (4, 21) = 3.55, *p* = 0.023, η2 = 0.40. After adjusting for covariates (age, sex, and RT), similar results were found. The univariate ANCOVAs showed that non-clinical burnout affected CE, F (4, 21) = 12.90, *p* = 0.001, η2 = 0.35, but not VRT [F (4, 21) = 0.003, *p* = 0.956], OE [F (4, 21) = 0.06, *p* = 0.811], or RT [F (4, 21) = 1.23, *p*= 0.279]. Analysis of covariance of VRT controlled for RTs, revealed a tendency for a significant group difference showing that, compared to the controls, the NCB group had higher VRT (*p* = 0.07, two tailed).

Considering CE as the dependent variable, Pearson correlation coefficients (R) reached statistical significance for the three burnout dimensions (Table 2): exhaustion (R = 0.545, *p* = 0.004); cynicism (R = 0.563, *p* = 0.003), and reduced personal effectiveness (R = −0.522, *p* = 0.006). It should be mentioned that we investigated all linear regression assumptions (plots of residuals against predicted values, histograms of residuals, and tests of normality) as well as Cook’s distances, DFBETAs, and Mahalanobis distances).

## 4. Discussion

Our data indicated that specific attention deficits occur in employees who are still on the job but experience burnout symptoms (non-clinical burnout). The performance on the attention test suggests that the behavior of the NCB participants tended to be guided by more impulsive responses than controls.

### 4.1. Executive Deficits in NCB in Healthcare Workers during the COVID-19 Pandemic

This study investigated NCB and attention performance deficits among healthcare workers during the COVID-19 pandemic, a context not extensively explored in existing literature. The pandemic’s unprecedented stressors provided a unique opportunity to study NCB in an intensified setting. Accordingly, in this study, the sample of NCB subjects manifested significant levels of exhaustion and depersonalization and a reduced sense of personal or professional accomplishments. In the context of our study, healthcare workers during the COVID-19 pandemic faced significant challenges in preserving their resources, such as emotional and physical energy, due to high stress, resource constraints, and increased workload. This depletion of resources might lead to decreased self-regulation, manifesting as increased impulsivity or hyperactivity in attention tasks. 

The basic paradigm of the CVAT task is characterized by a serial presentation of target and non-target stimuli, and the subject’s task is to respond to target stimuli and control inadequate responses [37,38,39,40,41]. Inhibition is the ability to control inadequate responses, measured by the number of CEs. The finding of a higher number of CEs suggests that NCB individuals may have difficulties with attention in daily tasks.

The significant difference between controls and NCB individuals in the impulsivity-hyperactive subdomain may indicate that these individuals are more prone to make mistakes during their working time. In this regard, it is acknowledged that sustained attention and response inhibition are aspects of executive control [64]. Executive control is a term that refers to a set of cognitive processes underlying voluntary regulation of perception and motor responses, to adaptively deal with changing task demands [65]. Individuals whose executive control is impaired typically show deficits in their ability to inhibit inappropriate motor responses [31]. Therefore, our data corroborates the hypothesis of executive dysfunction in the NCB individuals. However, the ability to sustain attention is also part of executive control. In this regard, it would be expected that NBC subjects would be impaired in the VRT variable. However, contrary to this expectation, VRT was not affected.

The absence of any significant difference in the VRT variable may be explained by the associations among VRT, RT, and CE. As indicated by the raw data (Figure 3), the average VRT did not differ between controls and NCB subjects. In contrast, NCB subjects tended to be faster than controls. However, there was a negative correlation between reaction time and inhibition errors. Analysis of covariance, in which we controlled for RTs, confirmed a significant effect for Group. Post-hoc ANCOVAs (group comparisons, controlling for RT) showed a significant group difference in CE and a tendency for significance in VRT. This tendency reflected that the VRT was higher in NCB subjects as compared to controls. In addition, previous studies [66] have demonstrated that VRT can be adequately measured using short-duration tests (e.g., 1-min test) provided that these tests have several measured reaction times, usually more than 25 trials. As the CVAT has 90 trials (72 correct targets) and lasts more than 1 min, the present finding of a higher VRT in NCB workers when the data were controlled for RTs corroborates our hypothesis of an executive attention problem in this population. 

Our findings support earlier clinical observations reporting that clinical burned-out individuals have difficulties with attention in daily tasks [2,7]. Linden et al. [7] administered the Sustained Attention Response Test (SART) and demonstrated that participants with clinical burnout performed significantly worse on CE and VRT. However, in an apparent contradiction to our finding, this study also reported that NCB and controls did not differ in SART performance. It should be mentioned that the SART has only 11% of non-targets whereas the CVAT has 20%. An increased number of non-targets in the CVAT increases the probability of identifying subjects who made incorrect responses due to impulsivity. Thus, it is possible that the CVAT would be more accurate than the SART to evaluate inhibition responses. In addition, it should be mentioned that Linden et al. [7] also reported that the level of executive attention deficits was highest in the clinical burnout group, intermediate in the high (non-clinical) burnout employees, and lowest in the control group.

To our knowledge, the current study is the first to describe empirical findings indicating that non-clinical burnout is associated with deficits in executive control. Recently, Koutsimani and Montgomery [27] reported visuospatial deficits in NCB subjects. Considering the pivotal role of attention in cognition, we speculate that these visual special deficits may, at least in part, reflect executive attention deficits.

### 4.2. Limitations

Although the current study describes important inhibitory deficits in NCB, there were also some limitations. 

First, the sample size was small, which limits the generalizability of the results and makes the statistical differences more susceptible to Type I error. However, the finding of a significant difference in CE with small groups suggests that the effects sizes of cognitive deficits associated with burnout may be substantial.

Secondly, burnout was examined through self-reported questionnaires, thus participants’ answers could be affected by potential self-report biases. Future studies should also be conducted using structured clinical interviews. 

Thirdly, we used only one test that measures one cognitive domain. Although the subject’s ability to attend to specific stimulus and inhibit inadequate responses (attention) must be established before the most complex functions are evaluated, future studies with larger sample sizes and using tests that assess other cognitive domains (memory, language, etc.) will allow a better comprehension of the influence of NCB in cognition. 

Finally, we did not control for the subjects’ stress levels, resources, self-regulation, and workload in this study. Although this is a cross-sectional study and does not allow any inference regarding causal effects, controlling the variables mentioned above would be helpful to investigate whether depletion of resources might lead to decreased self-regulation, manifesting as increased impulsivity or hyperactivity in attention tasks.

## 5. Conclusions

We showed that HCWs with high burnout levels who are still on the job performed badly on variables that are linked to specific attention subdomains (sustained attention and impulsivity). Our study suggests a way to objectively assess attentional impairments associated with burnout symptoms by using a brief computerized attention test alongside a self-reported burnout questionnaire. The current study provides evidence that using the CVAT with a self-reported burnout questionnaire could be used to identify individuals with high burnout symptoms and executive attention deficits. Adding this objective tool (CVAT) could help to develop new strategies to reduce accidents related to burnout at work, especially in cases mostly affected by impulsivity. Moreover, it would help policymakers measure burnout’s actual impact on working populations, particularly in countries with a high burnout prevalence, such as Brazil [67].

## Figures and Tables

**Figure 1 ijerph-21-00239-f001:**
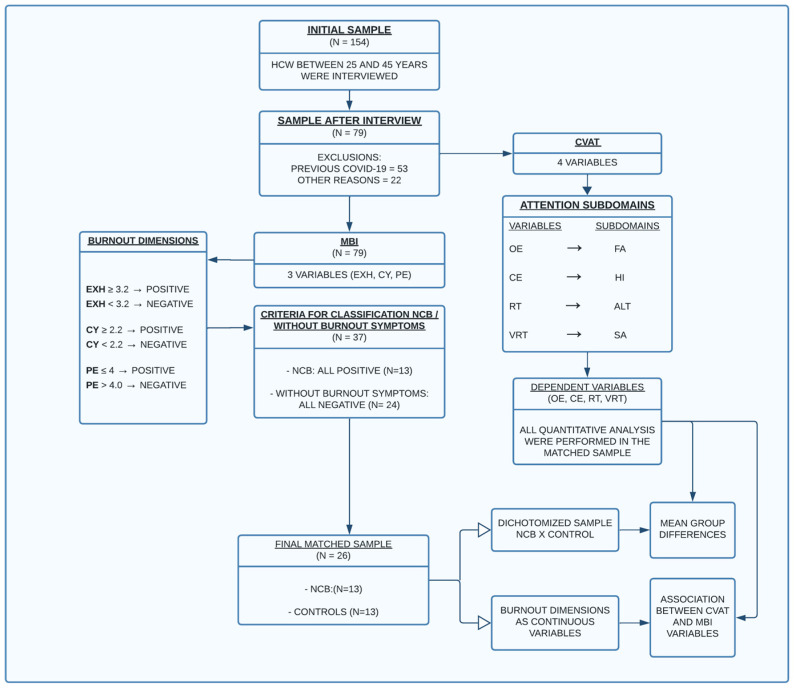
Flowchart explaining the study design. Abbreviations: CVAT, Computerized Visual Attention Test; MBI, Maslach Burnout Inventory; OE, omission errors; FA, focused attention; CE, commission errors; HI, hyperactivity/impulsivity factor; RT, reaction time; ALT, alertness; VRT, intraindividual reaction time variability; SA, sustained attention; EX = mean exhaustion; CY, mean cynicism; PE, mean personal efficacy; HCW, healthcare workers; NCB, non-clinical burnout.

**Figure 2 ijerph-21-00239-f002:**
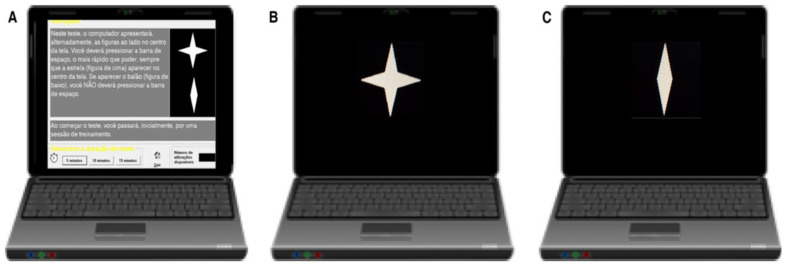
Schematic overview of the CVAT showing the target (star) and non-target (diamond). The CVAT begins with on-screen instructions (**A**): “In this test, the computer alternately displays the indicated figures in the center of the screen. You must press the spacebar using your dominant hand as fast as you can whenever the star appears in the center of the screen. If the other figure appears, you should not press the space bar”. Both the target (**B**) and the non-target (**C**) remained on the screen for 250 milliseconds (ms). The test consists of 90 trials, with either of the two figures presented in each trial. The interstimulus interval is 1 s, resulting in a total test duration of 1.5 min. Key variables include Omission Errors (OE), Commission Errors (CE), average Reaction Time of correct responses (RT), and Intraindividual Variability of Reaction Time (VRT, standard deviation of the RTs during the test). The CVAT [46] is available for research and clinical use by licensed psychologists. Requests for access can be made to the corresponding author Prof. Sergio L. Schmidt. There are versions in English, Spanish, and Portuguese. CVAT: Continuous Visual Attention Test.

**Figure 3 ijerph-21-00239-f003:**
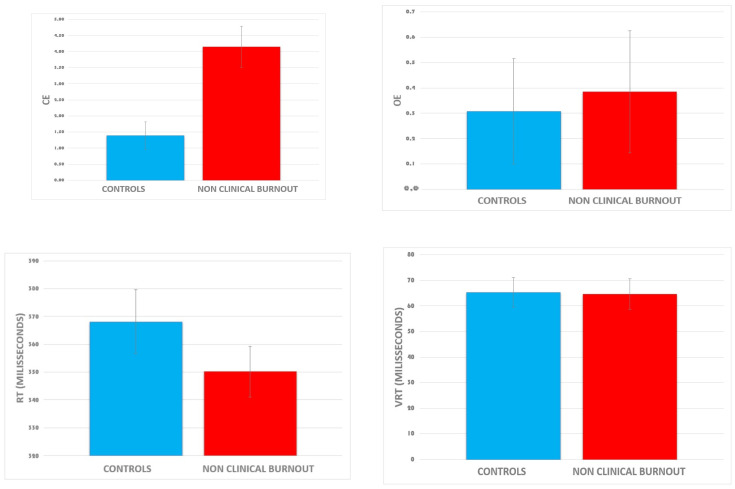
Means of each CVAT variable according to the group (raw data). Data are expressed as the mean ± standard error of the mean. Abbreviations: CE, commission errors; OE, omission errors; RT, reaction time; VRT, intraindividual reaction time variability. Group differences reached significance for the CE variable.

**Table 1 ijerph-21-00239-t001:** Demographics and MBI data (*n* = 26).

Demographics and Burnout Dimensions	Non-Clinical Burnout (*n* = 13)	Controls (*n* = 13)	Group DIFF (P)
FEMALE	5 (38.46%)	4 (30.76%)	ns
AGE	34.23 ± 7.0	36.38 ± 3.2	ns
PHYSICIANS	5 (38.46%)	6 (46.15%)	ns
NURSES AND NURSE AIDES	8 (61.15%)	7 (53.38%)	ns
**AVERAGE EXH**	**5.11 ± 0.59**	**0.73 ± 0.43**	**<0.01**
**AVERAGE CY**	**4.49 ± 1.12**	**0.14 ± 0.25**	**<0.01**
**AVERAGE PE**	**3.23 ± 0.32**	**5.53 ± 0.56**	**<0.01**

Each continuous variable is expressed as the mean ± standard deviation. Abbreviations: EXH, Exhaustion; CY, Cynicism; PE, Personal Efficacy; ns (non-significant); P, proof value. Significant group differences are indicated in BOLD.

**Table 2 ijerph-21-00239-t002:** Pearson correlation coefficients (*n* = 26).

Variables	Average EXH	Average CY	Average PE
CE			
R Pearson	0.545 *	0.563 *	−0.522 *
*p*-value	0.004	0.003	0.006
OE			
R Pearson	0.013	0.047	−0.079
*p*-value	0.950	0.818	0.703
RT			
R Pearson	−0.228	−0.318	0.334
*p*-value	0.262	0.114	0.095
VRT			
R Pearson	0.011	−0.106	0.074
*p*-value	0.957	0.608	0.720

The table includes correlation coefficients (R Pearson) and *p*-values for variables CE, OE, RT, and VRT against averages of EXH, CY, and PE. “R Pearson” refers to the Pearson correlation coefficient, which measures the strength and direction of the relationship between two variables. “*p*-value” refers to proof value. * Indicates that the result is significant at *p* < 0.01. Abbreviations: EXH, Exhaustion; CY, Cynicism; PE, Personal Efficacy; CE, commission errors; OE, omission errors; RT, reaction time; VRT, intraindividual reaction time variability.

## Data Availability

Data are available upon request to the senior author (S.L.S.—slschmidt@terra.com.br).

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
