# Peer review of "Attention Deficits in Healthcare Workers with Non-Clinical Burnout: An Exploratory Investigation"

_ijerph, 2024, doi:10.3390/ijerph21020239_

Round 1

Reviewer 1 Report (Previous Reviewer 2)

Comments and Suggestions for Authors

Well done!

Author Response

We are thankful to R#1 for the careful revisions. We are also glad to know that R#2 stated that the " An exploratory investigation deals with an important issue. The methodology of the study does not raise any objections, the authors examined the assumptions of the statistics used very carefully … thus should be published”.

The manuscript improved considerably after considering all the comments raised by the reviewers. We hope it will now merit publication in this prestigious journal.

Sincerely

Prof. Sergio L. Schmidt, M.D., Ph.D. - on behalf of all authors.

Reviewer 2 Report (New Reviewer)

Comments and Suggestions for Authors

The paper entitled: Attention deficits in healthcare workers with non-clinical

 burnout: An exploratory investigation deals with an important issue. The methodology of the study does not raise any objections, the authors examined the assumptions of the statistics used very carefully. The article extends our knowledge about burnout syndrome and cognitive deficits and thus should be published.

Small improvements are recommended:

1. The abstract includes unnecessary information e.g. information about stages of sample selection, as a result, readers are given the expression that 154 subjects not 26 took part in the study (13 from the clinical and 13 from the control group). It is also unnecessary to inform about the statistics used in the study. At the same time, it lacks information about the basic demographic characteristics of the sample and conclusion.

2. There are too many keywords, some of them refer to similar aspects but in a wider way e.g. cognitive deficits and attention performance

3. Introduction section includes numerous unreferenced sentences e.g. lines 48-51; 57-8; 78-80;86-91; 94-5; 103-6;412-13

4. Despite the fact, that the study is exploratory in nature, the authors hypothesized that “NCB subjects working in a stressful environment would show specific attention deficits, especially in the executive attention subdomain.” However, such a hypothesis needs justification and somehow is inconsistent with theories provided by the authors (The authors started their background by pointing to the Deligkaris et al. suggestions that cognitive deficits/impairments are the fourth components of burnout symptoms) and with the prior statements: “our study delves deeper into specific attention subdomains and how they affect an NCB population within a high-stress environment”. In summary, there is a logical inconsistency: once attention deficits are predictors of NCB, and once NCB leads to attention deficits. As a matter of fact, the way of experiment suggests the second way, as the authors selected NCB health workers and examined their attention performance.

5. In the result section the authors did not provide information about the profession of 5 subjects e.g. The sample of health professionals consisted of 11 physicians and 10 nursing professionals. (line 335-6).

6. Throughout the entire paper the authors several times used interchangeably terms “cognitive impairment” and “attention performance deficits”. For example in the discussion section, we can read: “This study investigated NCB and cognitive impairment among healthcare workers 391 during the COVID-19 pandemic (…). This is incorrect, as cognitive impairment is a term much wider than attention deficits, thus it should be changed.

7. Some descriptions in the discussion section are unjustified or unreferenced e.g. In the context of our study, healthcare workers during the 396 COVID-19 pandemic faced significant challenges in preserving their resources, such as emotional and physical energy, due to high stress, resource constraints, and increased workload. This depletion of resources might lead to decreased self-regulation, manifesting as increased impulsivity or hyperactivity in attention tasks. First, one can not say anything about the causal effects of a cross-sectional study. Next, The authors directly did not control subjects’ stress levels, resources, self-regulation, workload, etc.

8. Some references are incorrectly coded e.g. [e.g., 7]

Author Response

We are thankful to both reviewers for their careful revisions. R#1 considered the manuscript ready for publication. R#2 stated that the " An exploratory investigation deals with an important issue. The methodology of the study does not raise any objections, the authors examined the assumptions of the statistics used very carefully … thus should be published”. However, R#2 suggested eight minor corrections. All suggestions made by R#2 are in red in the revised version.

Q1R#2: “The abstract includes unnecessary information e.g. information about stages of sample selection … It is also unnecessary to inform about the statistics used in the study. At the same time, it lacks information about the basic demographic characteristics of the sample and conclusion.”

Answer Q2 R#2:

            Thanks. We made a careful revision and excluded all the unnecessary information raised by R#2. We also included information about the basic demography and a brief conclusion.  Please see the revised version of the abstract (the requested necessary information is in red).

Q2R#2: “There are too many keywords... ”

Answer Q2 R#2:

            We reduced the number of keywords.

Q3R#2: “Introduction section includes numerous unreferenced sentences.”

Answer Q3 R#2:

            We agree with R#2 and made the corrections. Please see lines 48,56, 78, 86, 89, 93, 104, 409

Q4R#2: “the authors hypothesized … such a hypothesis … is inconsistent … there is a logical inconsistency: once attention deficits are predictors of NCB, and once NCB leads to attention deficits. As a matter of fact, the way of experiment suggests the second way, as the authors selected NCB health workers and examined their attention performance.” 

Answer Q4 R#2:

We agree with R#2. Therefore, we made clear our hypothesis:  NCB leads to attention deficits. Please see lines 107-112

Q5R#2: “In the result section the authors did not provide information about the profession of 5 subjects e.g. The sample of health professionals consisted of 11 physicians and 10 nursing professionals.”

Answer Q5R#2:

  Corrected (lines 331-332)

Q6R#2: “…the authors several times used interchangeably terms “cognitive impairment” and “attention performance deficits”. For example in the discussion section, we can read: “This study investigated NCB and cognitive impairment among healthcare workers 391 during the COVID-19 pandemic (…).”

Answer Q6R#2:

            Ok. We changed “cognitive impairment” to “attention performance deficits” throughout the text when necessary. Please see lines 387

Q7R#2: “Some descriptions in the discussion section are unjustified … e.g. depletion of resources might lead to decreased self-regulation, manifesting as increased impulsivity or hyperactivity in attention tasks. First, one can not say anything about the causal effects of a cross-sectional study. Next, The authors directly did not control subjects’ stress levels, resources, self-regulation, workload, etc.”

Answer Q7R#2:

            We agree with R2 and removed unjustified descriptions. In the limitation subsection we stressed that this is a cross-sectional study and that we did not control subjects’ stress levels, resources, self-regulation, workload, etc. Please see lines 461-465

Q8 R#2: “Some references are incorrectly coded e.g. [e.g., 7]”

Answer Q8R#2:

            Ok. Please see lines 139 and 296

The manuscript improved considerably after considering all the comments raised by the reviewers. We hope it will now merit publication in this prestigious journal.

Sincerely

Prof. Sergio L. Schmidt, M.D., Ph.D. - on behalf of all authors.

This manuscript is a resubmission of an earlier submission. The following is a list of the peer review reports and author responses from that submission.

Round 1

Reviewer 1 Report

Comments and Suggestions for Authors

The authors did not collect more data to increase sample size and make a clear hypothesis in the revised version.

Comments on the Quality of English Language

No other comments. 

Reviewer 2 Report

Comments and Suggestions for Authors

Dear authors,

Your study is both interesting and important; however, there is room for improvement. Please consider the following suggestions:

  1. 1. Check the manuscript for typos and repetitions, particularly on page 3, lines 73-78. Ensure clarity and coherence in the text.

  2. 2. Justify why individuals older than 45 years of age were excluded. Provide a clear rationale for this exclusion criterion.

  3. 3. Explain why some words in the Methods section are bolded. Clarify the significance or criteria for bolding specific terms.

  4. 4. Clearly describe the study design in both the abstract and Methods section. Ensure that readers have a comprehensive understanding of the study's methodology.

  5. 5. On page 3, lines 135-136, it is mentioned, "All workers were interviewed at the beginning of their work shifts to exclude fatigue." Specify the number of workers interviewed as the first step.

  6. 6. Add the numbers of study participants to each of the boxes in Figure 1 to enhance clarity and completeness.

  7. 7. Consider reconstructing Table 2 instead of including a photo. Ensure that the table is presented in a format that is easily accessible and interpretable for readers.

  8. 8. Include a discussion of the study limitations. Address potential shortcomings and provide insights into how these limitations may have influenced the study outcomes.
